# Microbial Involvement in the Bioremediation of Total Petroleum Hydrocarbon Polluted Soils: Challenges and Perspectives

**Ilaria Chicca [1], Simone Becarelli [1,2] and Simona Di Gregorio [1,*]**

1   Department of Biology, University of Pisa, 56126 Pisa, Italy; ilaria.chicca@biologia.unipi.it (I.C.); simone.becarelli@biologia.unipi.it (S.B.)
2   BD Biodigressioni srl, 56126 Pisa, Italy
*   Correspondence: simona.digregorio@unipi.it

**Abstract:** Nowadays, soil contamination by total petroleum hydrocarbons is still one of the most widespread forms of contamination. Intervention technologies are consolidated; however, full-scale interventions turn out to be not sustainable. Sustainability is essential not only in terms of costs, but also in terms of restoration of the soil resilience. Bioremediation has the possibility to fill the gap of sustainability with proper knowledge. Bioremediation should be optimized by the exploitation of the recent "omic" approaches to the study of hydrocarburoclastic microbiomes. To reach the goal, an extensive and deep knowledge in the study of bacterial and fungal degradative pathways, their interactions within microbiomes and of microbiomes with the soil matrix has to be gained. "Omic" approaches permits to study both the culturable and the unculturable soil microbial communities active in degradation processes, offering the instruments to identify the key organisms responsible for soil contaminant depletion and restoration of soil resilience. Tools for the investigation of both microbial communities, their degradation pathways and their interaction, will be discussed, describing the dedicated genomic and metagenomic approaches, as well as the interpretative tools of the deriving data, that are exploitable for both optimizing bio-based approaches for the treatment of total petroleum hydrocarbon contaminated soils and for the correct scaling up of the technologies at the industrial scale.

**Keywords:** bioremediation; culturable and unculturable microbes; machine learning; metagenomic and genomic sequencing; predictive functional metagenomics



## 1. Introduction

In the era of renewable energies, crude oil is still one of the most important commodities in the world. Its uses span from energy generation and fuel for transportation to petrochemical productions such as plastics and solvents. Oil industry is one of the most powerful branches in the world economy, but it is based on finite reserves and, since an increasing number of conventional underground reserves are almost depleted, unconventional processes of oil extraction are nowadays feasible and consolidated. Examples are shale oil and tar sands, noxious for the environment, as well as the conventional oil drilling. United States and Canada are the major providers of shale oil and tar sands respectively, accounting for nearly one fourth of the world's crude oil production. Most of the rest of crude oil reserves are in the Middle East, housing the greatest segment of oil reservoirs [1]. The crude oil to consumers supply-chain consists in three segments. Potentially, the three segments damage the environment. The upstream one, comprises exploration and drilling activities. The midstream activities, comprise transportation and refinery. The downstream activities are represented by the supply chains of heating oil and fuels. Oil spillage, pipeline accident and vandalization are the main sources of contamination for the environment and they have already destroyed coastal marine and terrestrial areas, polluted ground and surface aquifers and led to geopolitical crisis. Moreover, crude oil processing is a source

of atmosphere pollution, interferences with ecological resources and release of hazardous materials with health and safety implications for the biosphere. A sustainable approach to the managing of noxious situations is mandatory and bio-based technologies may be an opportunity.

Crude oil is composed by different fractions, *n*-alkanes, aromatics, nitrogen-sulphur-oxygen compounds (NSO), resins and asphaltenes. The *n*-alkane or saturated fraction, ranging from $C_1$ (methane) to more than $C_{40}$, is composed by branched and not-branched chains of saturated hydrocarbons [2]. The aromatic fraction ranges from benzene, toluene, ethylbenzene and xylene (BTEX) or single-ring aromatic fractions, to multi-ring polycyclic aromatic hydrocarbons (PAH) or aromatic rings substituted with different alkyl groups [3]. Resins and asphaltenes have very complex and mostly unknown carbon structures with the addition of many nitrogen, sulfur and oxygen atoms [4,5].

Total petroleum hydrocarbon (TPH) is the term used to describe a large family of several hundreds of chemical compounds that originally come from crude oil. Since the chemical composition of crude oil is very complex, it is not practical to measure each component separately, but it is feasible, standardized and useful to measure and refer to the total amount of TPH when evaluating the level of contamination at a site. Most of the scientific literature on the biodegradation of crude oil refers to the TPH fraction, which includes both volatile and extractable petroleum hydrocarbons, encompassing the gasoline range organics ($>C_6$–$C_{10}$), diesel range organics ($>C_{11}$–$C_{28}$) and oil range organics ($C_{29}$–$C_{35}$). All these compounds are described as toxic to the environment.

The technologies dedicated to the protection of the environment or to the restoration of environmental quality, that have to face this complex and noxious scenario, must integrate the consolidated chemical-physical and less consolidated bio-based technologies to reach significant goals in terms of success. Bio-based technologies can be more sustainable in terms of costs because they are less energy-intensive. They are also eco-friendly since they are based on exploiting the potential of ecosystems to recover resilience in the various ecological niches. Here we review what is known, from a biological point of view, on the metabolic potential of microorganisms towards TPH, and which instruments we have to increase our capacity to exploit their potentiality, focusing on the adoption of "omics" in the environmental protection sector, with a focus on soil, one of the main ecosphere compartments affected by crude oil contamination [6].

## 2. The Soil Scenario

Soil crude oil and TPH contaminations are mainly consisting of weathered ones, except for fresh accidental spills. Indeed, processes of adsorption, photo transformation, biological transformation and volatilization of the diverse crude oil and TPH components are at the base of the transformation of fresh contamination in weathered ones in soil [7]. The process of weathering chemical structures in the environment is strictly dependent on the complexity of their chemical structures, which determine potential volatilization, water solubility, and loss of bioavailability [8]. In soil, the weathering of TPH is principally consistent with their sorption to the soil organic matter (SOM). The SOM in terms of amount [9] and nature [10] is mainly responsible for the decrease of bioavailability of TPH in soil. In fact, partitioning processes of TPH in the SOM entrap the contamination in soil micropores [11,12]. Once the contamination is entrapped in the SOM, the establishment of diverse processes of interaction among chemical structures with similar moieties (e.g., humic fraction of the SOM and organic contaminants with aromatic moieties) occurs. These interactions consist in dipole–dipole, dipole-induced dipole and hydrogen bonding [13]. Moreover, interactions between TPH and the soil mineral constituents also occur [14–16], strengthening the decrease in TPH bioavailability in soil, especially with the aging of the contamination. The higher is the molecular weight of the organic contaminants, the higher the number of aromatic moieties and functional groups capable to interact with the SOM. Consequently, a weathered oil-contaminated soil is usually dominated by high molecular weight hydrocarbons, significantly recalcitrant to biodegradation [17,18]. In this context,

the high molecular weight hydrocarbons are actually less toxic than the low molecular one, since these latter ones are less bioavailable for plants and microorganisms. The lag phase of bacterial growth was described as longer in the presence of higher concentrations of bioavailable hydrocarbons, and plant growth is inhibited by their presence since their hydrophobicity decreases the soil capacity to absorb water and nutrients [7].

On the other hand, it is generally accepted that microbes can eventually transform all the surrounding organic material over time. In particular, contaminants in environmental matrices exert a selective pressure on microbial communities. Speciation processes eventually occur and with time microbes, develop strategies to transform contaminants. Historically contaminated matrices are a useful reservoir of these organisms. Microbial strategies for contaminant transformation include not only degradative pathways but also the increment in their bioavailability and even their immobilization to contrast their toxicity (e.g., polycondensation or humification of the organic matter). Consequently, it is not surprising that the persistence of the TPH fractions in the soil is associated with the speciation of microbial strains that develop the capacity to transform the fraction with time [19].

Microbial biodegradation and biotransformation of organics vary in accordance with their chemical structures and the functional capabilities of the microbial community, specialized for the transformation of the parent molecule and of its intermediates of degradation. These latter ones are actually significantly important for the fate of pollutants in the environment, since in some cases, degradation products of the primary pollutants are more prone to subsequent transformations than the parent contaminant and, therefore, may be mineralized [20]. Other degradation products, such as polyaromatic chemical structures, are more susceptible to interaction with the SOM [21]. At the same time, the microbial catabolism is driven by pollutant concentration, since the activation of the microbial-degrading metabolisms needs a minimum threshold of pollutant concentration [22]. To reach these threshold levels, the bioavailability or the water solubility of the contaminant is mandatory. Consequently, in soil, TPH biodegradation rates are based also on the sorbed phase of the contamination [23–27]. Indeed, microorganisms reach the sorbed phase of the contamination because of their capacity to produce surfactants, responsible for both the increase in contaminant bioavailability and for facilitating the fixation of microorganisms onto the surface of the sorbed phase contaminants [28,29]. Thus, it is evident that there are a plethora of chemical-physical and biological elements that determine the destiny of organic pollutants in soil, and the condition is complicated by the complexity of the composition of the contamination.

## 3. Hydrocarburoclastic Bacterial Pathways

The aerobic bacterial metabolic pathways involved in TPH transformation are mainly intracellular. Regarding the aromatic fraction of TPH, the mechanisms developed by bacterial cells to assimilate aromatic compounds have been fixed and optimized by natural selection, giving rise to functionally catabolic pathways organized in a funnel-like topology. In fact, a wide diversity of aromatics is directed via different peripheral pathways to a few key central intermediates. The latter are subjected to de-aromatization and further conversion to intermediary metabolites, such as acetyl-CoA, succinyl-CoA or pyruvate, via central pathways [30]. Hydroxylating oxygenases and ring-cleavage dioxygenases are responsible, respectively, for the hydroxylation and oxygenolytic cleavage of the aromatic rings. The products of these oxidative passages converge both on cathecolic structures that are subjected to ortho or meta cleavage by intradiol or extradiol (type I and II) dioxygenases, respectively, and on non-catecholic structures such as gentisate, homogentisate, monohydroxylated aromatic acids and heteroaromatic flavonols, that are subject of ring cleavage by dedicated type III extradiol dioxygenases [31]. Moreover, dioxygenases such as the CO-forming 1-H-3-hydroxy-4-oxoquinaldine 2,4-dioxygenase (HOD) and 1-H-3-hydroxy-4-oxoquinoline 2,4-dioxygenase (QDO) are involved in the transformation of N-heteroaromatic compounds [32]. In Figure 1, the scheme of the main biochemical strate-

gies to degrade aromatic structure (benzoate) in aerobic bacteria is reported. The aerobic hybrid pathway (Scheme B, Figure 1) shares the same activation reaction involved in anaerobic degradation.

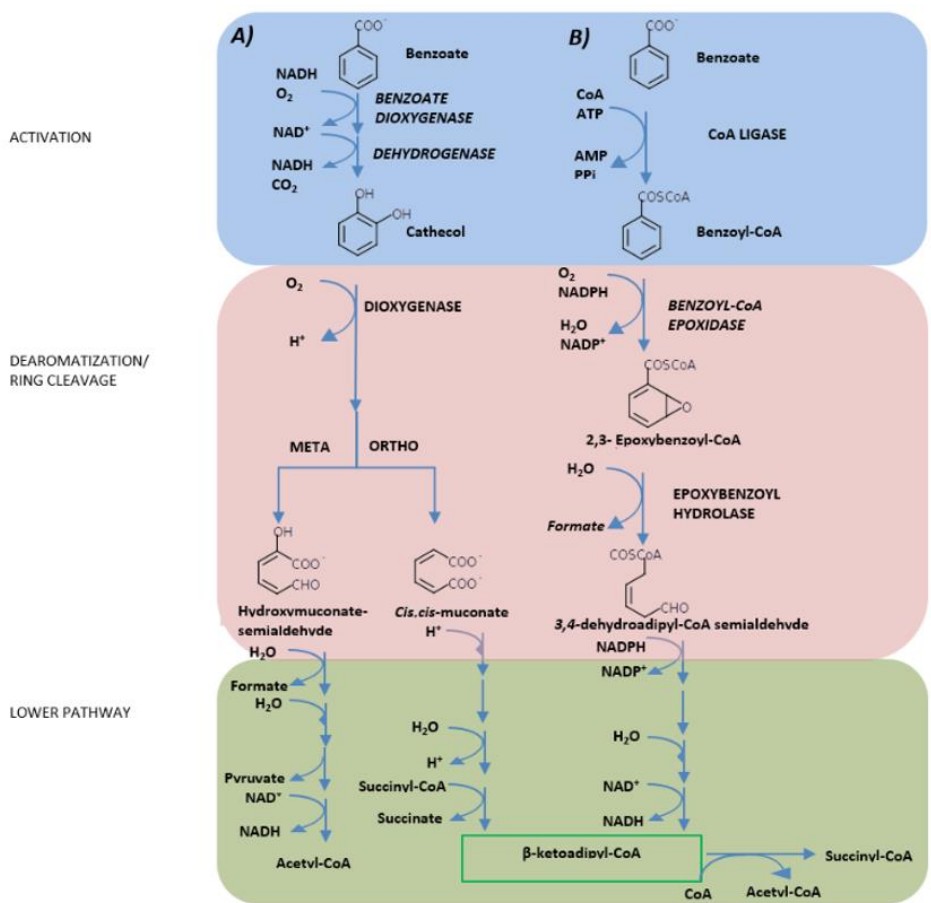

**Figure 1.** Benzoate can be aerobically catabolized following two major strategies: (**A**) classical aerobic biodegradation pathway and (**B**) aerobic hybrid pathway. In both strategies, an activation step (blue), dearomatization/ring-cleavage step (pink) and further degradation to central metabolites, that is, lower pathway step (green), can be identified. The ortho cleavage of catechol (b-ketoadipate central pathway) and the benzoyl-CoA hybrid pathway converge into the common b-ketoadipyl-CoA intermediate. The anaerobic degradation of benzoate shares a similar initial reaction with the aerobic hybrid pathway catalyzed by a benzoate-CoA ligase.

Regarding the TPH saturated fraction, this is composed by *n*-alkane whose oxidation is intracellular and initiated by oxygenases that introduce oxygen atoms into *n*-alkanes by four different pathways (Figure 2). The terminal oxidation pathway [33] is involved in the oxidation of the *n*-alkanes terminal methyl group. The product of the reaction is a primary alcohol, further oxidized by alcohol dehydrogenases and aldehyde dehydrogenases to fatty acid that enters in β-oxidation [34]. The termini of the *n*-alkane can be oxidated to the corresponding fatty acid without breaking the carbon chain by the biterminal oxidation pathway. The product of the reaction is a ω-hydroxy fatty acid, further converted to a dicarboxylic acid, entering in β-oxidation [34–36]. Subterminal oxidation has been also observed to form primary alcohols and secondary alcohols or methyl acetone with the same chain length as the substrate [37]. In *Acinetobacter* sp. strain HO1-N [38], the *n*-alkanes are oxidized to form *n*-alkyl hydroperoxides and then peroxy acids, alkyl aldehydes and finally, fatty acids. A dioxygenase is responsible for the first step of oxidation [39].

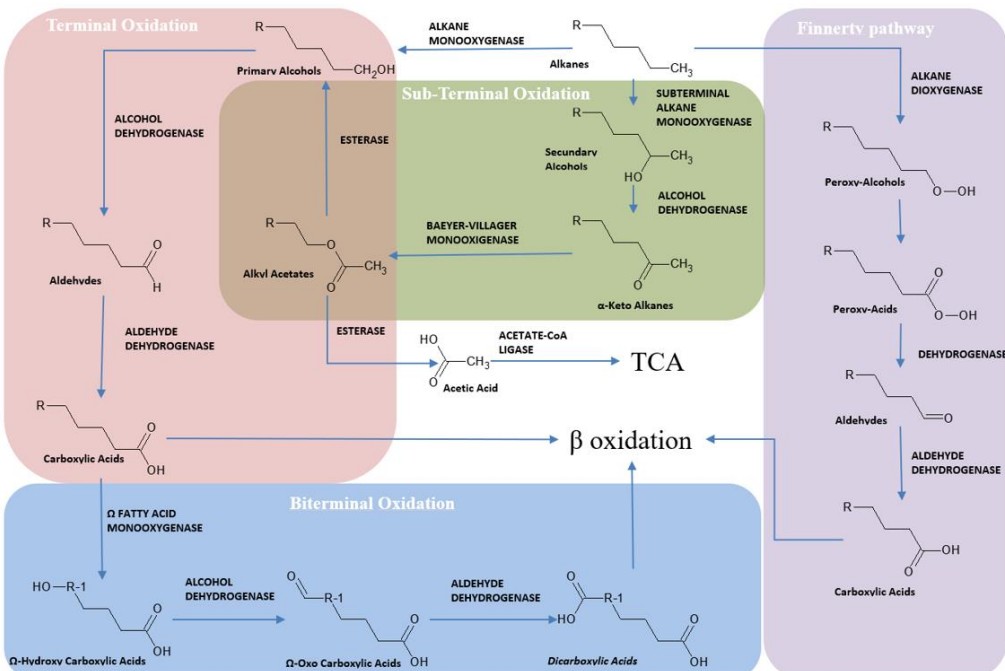

**Figure 2.** Possible branches of the aerobic degradation of *n*-alkanes. A bifurcation is possible at the end of terminal oxidation pathway: the obtained carboxylic acid can ether go through β-oxidation or be further oxidized by the ω-fatty acid mono-oxygenases to form dicarboxylic acid (biterminal oxidation). The products of Subterminal oxidation pathway are secondary alcohols or methyl acetone, which can be further oxidized by Baeyer–Villiger mono-oxygenases and esterases to generate fatty acids and primary alcohols. The first step of Finnerty pathway is the formation of n-alkyl hydroperoxides by alkane dioxygenases that are oxidated to fatty acids.

The water solubility and consequent bioavailability of both the aromatic and saturated fraction of TPH are low. The latter is eventually increased by the capability of most of the hydrocarburoclastic bacteria to produce biosurfactants. Biosurfactants are of diverse chemical compositions such as glycolipids, fatty acids, lipopeptides and lipoproteins, phospholipids and neutral lipids [40]. Biosurfactants are amphiphilic molecules with alkyl chains linked to sugar molecules, resulting in hydrophobic and hydrophilic regions, respectively [41], reducing the surface tension at the water/oil interface, leading to emulsification of hydrophobic moieties [42]. Indeed, biosurfactants enhance the bioavailability of contaminants for microbial degradation, by improving the solubilization of hydrocarbons in water layers, where bacterial metabolism and spreading occur [43]. Hydrocarburoclastic bacteria have been described as capable to produce biosurfactants in situ, which promote their survival in hydrophobic compound-dominated environments [44]. Different bacterial genera are described as capable to produce biosurfactants, among others, *Pseudomonas*, *Bacillus*, *Acinetobacter*, *Alcaligenes*, *Rhodococcus* and *Corynebacterium* spp. [40,45,46]. The exploitation of hydrocarburoclastic bacteria producing biosurfactants find application in the acceleration of the bioremediation of polluted soil and sediments [46–48], even though fragmented information are available on chemical and physical properties of biosurfactants produced by hydrocarburoclastic bacteria during the hydrocarbon degradation process [49].

## 4. Fungal Pathways

Fungi are very competitive in disturbed ecological niches since they are able to spread in the environment via hyphae elongation and adopt growth strategies to resist physical stresses, such as lack of nutrients and water by osmo- and xero-tolerance [50–52]. Fungi have been described also as more efficient than bacteria in the degradation of high molecular weight hydrocarbons in soils [53,54], due to their capacity to secrete extracellular a-specific polyphenols oxidases and laccases that are capable to transform macromolecules recalci-

trant to biodegradation, such as lignin and soil organic matter [55,56]; these latter ones show structural similarities to the TPH aromatic fraction [57,58]. The reaction catalyzed by these enzymes is the a-specific extraction of electrons from polyphenolic macromolecules to yield radical cation intermediates. These intermediates are channeled to the aromatic ring opening, with the breakdown of phenolic and aromatic compounds, or to polycondensation reactions [57]. In soil, the polycondensation reaction of organic compounds is responsible for the synthesis of the SOM [59]; indeed, microorganisms catalyze the stabilization of the organic matter by polycondensation (humification) reactions [60]. In contaminated soils, the organic portion consists in part or mainly in the contamination; consequently, it is reasonable to assess that microbial specimens catalyze the stabilization of the bioavailable portion of the contamination by its humification. Basidiomycetes have been extensively described as ligninolytic fungi and are consequently capable to produce the extracellular polyphenols oxidases and laccases previously described. However, Ascomycetes have been found to be dominant in TPH-contaminated soils [61]. Numerous studies have demonstrated the ability of Ascomycetes to transform recalcitrant compounds [46,62–64], as well as their involvement in the synthesis of soil organic matter and their ability to catalyze extracellular polymerization of polyphenols [65]. On the other hand, Ascomycetes have been described for the involvement of also cytochrome P450 monooxygenases (CYPs) in the oxidation of aromatic structures [56,61,66]. The fungal CYP biodegradation pathway of the polyaromatic structures consists of initial oxidation of the aromatic ring and conversion in hydroxy, dihydroxy, dihydrodiol and quinone derivatives [53]. Oxidized metabolites are conjugated and stored in cellular organelles and lipid-vesicles [67,68], or secreted in a more soluble and biodegradable form [69,70] (Figure 3). Fungal Cytochrome P450, by catalyzing oxidation of hydrocarbon C–H bonds to the corresponding hydroxy (C–OH) products, are responsible also for the initial hydroxylation of n-alkane [71–75]. Similarly, to bacterial metabolism, the alcohol is oxidized to the corresponding aldehyde and then to the corresponding fatty acid [54]. Fungal di-terminal and subterminal oxidation have been also observed [2,54,76,77]. Indeed, fungi, in relation to the aromatic and saturated fraction of TPH adopt both intracellular and extracellular strategies of degradation.

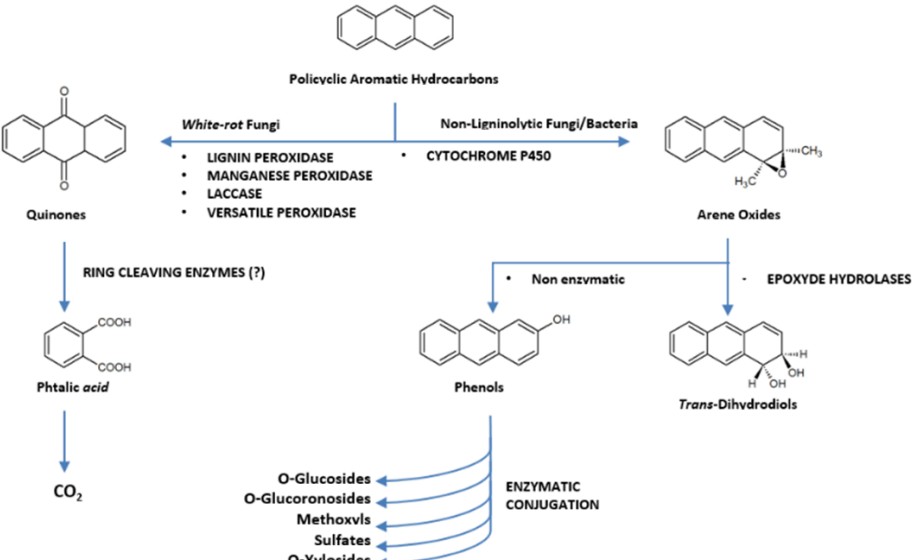

**Figure 3.** General pathways and enzymes involved in ring cleavage and/or oxidation of poly-cyclic aromatic hydrocarbons.

## 5. Microbial Interactions

Synergisms between microorganisms in the environment might be explained by the generalization of the ecological concept of K-r strategy, which basically explains two different approaches of nutrition of "organisms" sharing the same niche. The r-strategy consists

of high growth rates with high availability of carbon sources, while K-strategy consists of slow rates of growth and low availability of carbon sources. K-strategists are favored under nitrogen limitation since K-strategists are able to decompose the SOM, recalcitrant to biodegradation, for mineral nitrogen acquisition [78,79]. In parallel, r-strategists commensally utilize the SOM-derived compounds solubilized by the K-strategists. The r-strategists are favored in presence of available carbon and higher nitrogen availability [80]. A significant shift in microbial community structure from K-strategists to r-strategists is observed with the increased availability of nitrogen and carbon [81]. The shift might be observed also in terms of fungal abundances (K-strategists) in favor of bacterial ones (r-strategists), in consequence of the capacity of fungi to transform the SOM. It might be possible that the real situation is more complicated but it is evident that fungi and bacteria share microhabitats, assembled into dynamic co-evolving communities, described in almost every ecosystem [82]. These communities include microbial species from a wide diversity of fungal and bacterial families [83]. Microbial interactions contribute to soil functions as well as, and eventually even more than, species diversity [84]. Generally, the co-occurrence and synergism between fungi and bacteria in the soil might be based on the bacterial capacity to utilize fungal-secreted metabolites and overcome fungal defense mechanisms [85]. On the other hand, an example of the synergisms between the two kingdoms is the one of fungal lignocellulose decomposers, *Clitocybe* and *Mycena* spp., that interact with potential $N_2$-fixing bacteria, responsible for nitrogen deposition in soil during the decay of leaves [86,87]. Bacteria may contribute to nitrogen nutrition of fungi while fungi produce carbon sources for bacteria. Moreover, in oligotrophic habitat, fungi promote bacterial growth by nutrient and water transfer from fungal hyphae to the bacterial cells [50]. Model simulations show that fungal hyphae are spreading bacteria towards a source of contamination in environments where the active movement of bacteria towards pollutant reservoirs is limited by physical barriers [88–90]. In the context of crude oil bioremediation, it is worth mentioning that fungal hyphae have been described to mobilize PAHs by entrapping and transporting the latter in cytoplasmic vesicles [91], providing entrapped PAHs to hydrocarburoclastic bacteria [92]. In laboratory models of water-unsaturated environments, fungi transport the contamination from the water-depleted niches to the less-dried ones, where bacteria are blooming [93]. Moreover, fungi excrete organic molecules such as organic acids or polyols that activate bacterial chemotaxis towards the hyphae [94,95]. Indeed, fungi promote ecosystem functioning in heterogeneous habitats by transporting resources from high nutrient and water levels to nutrient-poor and dry areas.

Other than chemotaxis between fungi and bacteria, another useful chemical signal tool in the ecology is quorum sensing (QS). Quorum sensing enables bacteria to communicate by exchanging signaling molecules also referred to as "autoinducers". Two types of autoinducers are described: intraspecific signals, which are used by the same organism within a population and interspecific signals, which are used for communication among the entire microbial community [96]. A strong correlation between the expression of genes involved in hydrocarbon depletion and quorum sensing has been observed; as an example, a positive correlation between quorum sensing and the expression of the 2,3-catechol dioxygenase of the meta-cleavage (lower) pathway for hydrocarbon degradation has been reported [97]. On the other hand, the same correlation has been observed for quorum sensing modulation of the level of transcription of dioxygenase genes in the upper BTEX oxidation pathway [98]. QS is involved also in fungal morphogenesis, fungal biofilm development, apoptosis and eventually, pathogenicity [99]. Moreover, QS has been described as involved in interkingdom signaling in mixed fungal-bacterial biofilms [100–103].

## 6. Plasticity of Microbial Metabolic Profiles

Among bio-based approaches, mycoremediation has achieved a significant number of reasonable successes, especially in the case of bioaugmentation of indigenous fungi to soils and sediments contaminated by TPH [46,104]. In cases where bacterial ecology

has been studied during the contaminant degradation process, a significant effect of the bioaugmentation of fungal strain on the latter has been observed [46,104]. Bioaugmentation of autochthonous fungi to aged contamination has been observed as capable to promote the establishment of active hydrocarbon-degrading bacterial populations, competent for the degradation of both the aliphatic and the aromatic hydrocarbons [104]. On the other hand, the bioaugmentation of an autochthonous Ascomycetes to a TPH-contaminated soil, a *Ciboria* sp. strain, accelerated the onset of specialist bacterial species, competent for the transformation of the aromatic fraction of the contamination [46]. In this study, a functional predictive metabarcoding analysis of the bacterial ecology was adopted, and bacterial genera such as *Arthrobacter*, *Dietzia*, *Brachybacerium*, *Brevibacterium*, *Gordonia*, *Leucobacter*, *Lysobacter* and *Agrobacterium* spp. were identified as generalist saprophytes, essential for the onset of hydrocarbonoclastic specialist bacterial species, identified as *Streptomyces*, *Nocardoides*, *Pseudonocardia*, *Solirubrobacter*, *Parvibaculum*, *Rhodanobacter*, *Luteiomonas*, *Planomicrobium* and *Bacillus* spp. The functional traits that resulted to be indicative of the functional diverse roles of the two groups of bacterial species was the Dye decolorizing peroxidases (DyP). DyP has been retrieved both in eukaryotic and prokaryotic organisms, showing a higher diversity in prokaryotes [105]. Bacterial DyP may have high redox potentials, capable of oxidizing phenolic [106] and non-phenolic lignin model compounds [107]. The DyP, in the case study, was associated with the saprophytic metabolisms of the harboring bacterial genera, more than their capacity to deplete the contamination. As a matter of the fact, the saprophytic metabolism of the bioaugmented *Ciboria* sp. was positively synergizing with the bacterial one, accelerating the onset of bacterial specialists for the degradation of the aromatic fraction of the contamination, eventually accelerating its depletion. In this context, it should be mentioned that the saprophytic metabolisms of fungi and bacteria might be responsible for the transformation process of the organic matter in the soil, determining both the mobilization of carbon source via a partial transformation of the SOM and the stabilization of the latter, by eliciting polycondensation reactions. On the other hand, it is reasonable to assess that the aging of the contamination might render the polluted soil an oligotrophic environment, due to the lack of bioavailability of the dominant carbon source due to weathering processes. Microbial saprophytic metabolisms might be pivotal to the increase in bioavailability of the contaminants, deriving from the saprophytic partial oxidation of the main carbon sources in contaminated soils, that is consisting with the SOM-sorbed contamination. Thus, synergizing saprophytic metabolisms of fungi and bacteria might be pivotal to prime the actual depletion of the SOM-sorbed contaminants by both direct mechanisms (stabilization) and an indirect one, activating the metabolisms of specialist species, recalling the ecological concept of K-r strategy.

As previously assessed, a saprophytic metabolism mediated by microbial peroxidases is responsible for the humification (stabilization) of the organic matter in the soil. The humification of organic matter consists of a process defined as composting. Composting is an aerobic oxidative process that relies on the actions of microorganisms to degrade organic materials, resulting in the thermogenesis and production of organic and inorganic compounds, and on humification of the organic matter with a consequent stabilization. Composting strategies in biodegradation/bioremediation of organic pollutants have been seriously adopted for the last 20 years. In relation to the application of the protocol to soil remediation, there are a variety of composting systems, mainly consisting of more or less controlled and engineered windrows and open piles. Initially, many of these systems were developed for the stabilization of sewage sludges, catalyzing processes of stabilization, or humification of the corresponding organic fraction. With reference to the scenario of interest in the context of TPH contamination of soil, it is reasonable to assess that, where pollutants are completely bioavailable and biodegradable, composting processes should be favorable. On the other hand, the limited bioavailability, eventually due to aging of the contamination, might be accompanied by processes of stabilization of the contamination, determining a depletion of the contamination in terms of measurable parental components [108]. This assessment is corroborated by the evidence that during

composting of high molecular weight hydrocarbons, their mineralization, quantified by the development of $CO_2$ during the soil composting process, is limited and inversely correlated to depletion of the contamination [109], suggesting the occurring of stabilization processes more than mineralization of highly recalcitrant to biodegradation pollutants. This aspect, if inducing perplexity from the toxicological point of view [48], is advantageous for the optimization of bio-based processes. In fact, if saprophytic microorganisms are involved in increasing contaminant bioavailability and eventually in determining the conditions for the onset of microorganisms capable of transformation of the contamination (specialist species), but also in processes of stabilization of the contamination, the bioaugmentation approach to bioremediation might be re-designed by the exploitation not only of the specialist species for contaminant transformation but also of the saprophytic specimen showing resistance to the contamination. In a study concerning the exploitation of co-composting of lignocellulosic material and contaminated soils, it was observed that a wide variety of microorganisms (bacteria and fungi) are involved in the biodegradation of TPH. Fungi were descibed as producing enzymes for a partial oxidation of TPH with bacteria utilizing these partially oxidized products as carbon and energy sources. The bacterial ecology was described as changing during the successive phases of the composting process. The fungal ecology was described as mainly associated or deriving from the bulking lignocellulosic material, resulting as microbial effectors priming both the whole composting process and the interaction with the bacterial ecology, at the base of the efficiency of the biodegradative microbiome [110].

In another study, it was observed that the addition of bulking agents, such as rice straw or sawdust, improved the contaminant degradation rate. With a metagenomic analysis on fungal and bacterial communities, the authors concluded that the removal efficiency was related to a selective effect of the bulking agents on specific microbial communities. The community composition analysis shows that the abundance of petroleum-degrading taxa such as the bacterial *Sphingomonas* and *Phenylobacterium* spp. and the fungal *Humicola* and *Graphium* spp. increased synergistically after addition of rice straw and sawdust, with a significant acceleration of TPH depletion over time [111].

As a matter of the fact, the interaction of bacteria and fungi seems to be pivotal for the recovery of polluted soils and widely diversified microbial metabolic pathways, characterizing diverse prokaryotic and eukaryotic microbial species which actively sustain the decontamination process. In fact, it is already accepted that the degradation of recalcitrant and polluting organic chemical structures, in a complex matrix like the soil, cannot be achieved individually, but by complex community interactions using different metabolic strategies of microorganisms adapted to the latter [19]. However, many of the bio-based processes dedicated to the decontamination of polluted environments are basically limited to the isolation, characterization and bioaugmentation of specialist species. As previously assessed, it is not the single specialist, but microbial assemblages with syntrophic associations that are principally responsible for the transformation of contaminants in the environment. Moreover, in bioremediation, the depletion of the contamination must be accompanied by the "re-shaping" of the soil as a substrate offering ecosystem services [112,113]. In this context, the involvement of plant growth-promoting (PGP) bacteria that contribute to the restoration of soil quality, by their interaction with the primary producers colonizing the soil, might be of interest since it has also been described as involved in the degradation of soil contaminants, both directly and indirectly, improving plant performances in phytoremediation [114]. Microbial isolates combining both xenobiotic degradation capacity and PGP potentials might be an efficient and sustainable tool to be exploited in the planning of bio-based remediation interventions for the restoration of contaminated soils [115]. The design of synthetic microbial microbiota for biaugmentation, combining degradative and re-shaping functions, is desirable.

## 7. Taxonomical and Functional Metagenomics for Bioremediation

The updated scenario of the microbial processes of TPH transformation in contaminated soils reveals levels of complexity that risk to associate the exploitation of bio-based processes to uncertainty. In order to transform the bio-based approaches to soil decontamination in robust technologies, a comprehensive understanding of the physiology, ecology and phylogeny of the colonizing microbial consortia is mandatory. The capacity to isolate microbial specialists for the biodegradation of TPH and crude oils is crucial for evaluating the biodegradation process they catalyze, as well as the regulatory mechanisms that influence their activities in contaminated soils. Moreover, the sequencing of their genomes is mandatory to deposit sequences of interest for in silico analysis of all the soil matrices contaminated by the same class of contaminants. Nowadays, the application of "omics" techniques in the investigation of pure cultures gives the opportunity to assess the genetic diversity of specialist species and of all environmental microbes, to investigate genes responsible for contaminant degradation and their regulation. In this context, it should be mentioned that "omics" investigation can be exploited for both isolates and environmental samples [116–118] and the adoption of metagenomics, transcriptomics and proteomics offers the opportunity to design new approaches to the management of processes dedicated to the restoration of the environment [119,120]. These techniques can be exploited to study the organization, interdependence, physiology, ecology and phylogeny of environmental microbiomes, comprising the hydrocarbuclastic ones. The study of microbiomes cannot be limited to culturomic approaches. Indeed, the large majority of microorganisms colonizing natural habitats are either uncultivable or extremely difficult to cultivate. Metagenomics is designed to study these microbes [121], with the recovering of DNA sequences from environmental DNA, offering the opportunity to discover genes and biodegradative pathways [122,123]. The correspondence between the kinetics of decontamination and the variations of the microbial ecology is mandatory to assess the microbial populations that are involved in the decontamination of TPH-contaminated soils [48]. The predictive functional metagenomics have been exploited to dissect bacterial functions that are dominant in processes of TPH depletion in soils, synergically catalyzed by fungi and saprophytic bacteria, introducing possible mechanisms of interaction between the fungal and the bacterial hydrocarburoclastic community in an aged TPH soil contamination [48].

The extremely useful functional metagenomic analysis can be successfully based also on the cloning of environmental DNA fragments in selected microbial hosts, successively screened for environmental functions of interest. The sequencing of the fragments cloned in the selected hosts leads to the sequencing of gene coding for these functions [124], enriching environmental databases on pollutant-degrading genomic sequences of known and unknown, isolated and not yet isolated, microbial candidates, facilitating in silico analysis of all the environmental matrices, affected by the same contamination. The described functional analysis combined with taxonomic metabarcoding and the derived predictive functional analysis should be considered as the bases for developing a robust predictive instrument to infer the functions that the microbial communities and colonizing contaminated matrices can express and, consequently, the functions to be exploited to complete and even accelerate a decontamination process. Due to the availability of a plethora of metagenomic, in silico functional and wet-lab functional data, bioinformatic processing applications for environmental DNA sequence-based screening of metagenomes are extremely important and they tend to be adopted among other methodological approaches to design bio-based technologies for the recovery of contaminated matrices. Among others, a list of the most important, frequently utilized and open-source pipelines of analysis follows.

Meta Genome Analyzer (MEGAN) analyzes large amounts of metagenomic sequence data [125] comparing metagenomic and metatranscriptomic data both functionally and taxonomically, mapping reads to the NCBI, SEED, COG and KEGG classificators. The tool is robust and user-friendly.

Community Cyberinfrastucture for Advanced Microbial Ecology Research and Analysis (CAMERA) is a database and computational tools providing the possibility for depositing, locating, analyzing and sharing data about microbial biology [126,127]. CAMERA offers the opportunity to deposit and work with metadata relevant to environmental metagenome datasets with annotations in a semantically aware environment. The user can use semantic queries to search in the database. CAMERA might be defined as a complete genome-analysis tool allowing users to analyze both metagenome and genome data.

Subsystem Technology for metagenomes (MG-RAST) is an automated platform providing quantitative insight into microbiomes based on their sequence data [128]. The pipeline performs similarity-based annotation on nucleic acid datasets, clustering and protein prediction, phylogenetic and metabolic reconstructions of genomes and metagenomes.

Phylogenetic metabarcoding is fundamental to study the microbial ecologies of any environmental niches. However, the molecular tool does not provide direct evidence of the community functional capabilities. PICRUSt (phylogenetic investigation of communities by reconstruction of unobserved states), now PICRUSt2 [129], offers a computational approach to predict the functional composition of a metagenome using marker gene data and a database of reference genomes. PICRUSt uses an extended ancestral-state reconstruction algorithm to predict which gene families are present and then combines gene families to estimate the composite metagenome. Using phylogenetic information, PICRUSt2 infers key findings from the Human Microbiome Project and accurately predicts the abundance of gene families in host-associated and environmental communities, with quantifiable uncertainty.

Indeed, metagenomics is fundamental to understanding the composition and the ecology of hydrocarburoclastic microbiota that can be defined as bucket-brigades for the degradation of TPH and crude oils in soils. Their functional analysis is fundamental to recover hydrocarburoclastic microbiomes, combining taxonomy and functional information, providing the instruments to deepen the information provided by a conceptual model of a contaminated site; in fact, the latter can comprise not only data about the stratigraphy, geological characteristics and chemical profiles of the contamination in the contaminated site. Indeed, the site will be characterized also in terms of the colonizing microbial communities, whose functional traits will be automatically inferred by the exploitation of data provided by the constantly increasing number of metagenomic studies. In this direction, it is worth mentioning that the predictive metagenomic profiling successfully described phylogenetic and functional compositions of diverse oil-polluted sites around the world, allowing the inferring of metagenomic features including taxonomical markers and functional modules, to be used as biomarkers for the effective distinction between diverse oil-polluted sites [130].

## 8. Conclusions

Biodegradation is defined as a sustainable approach to the decontamination of soils, however, for the transferability of the technology onto the industrial scale, repeatability, robustness and predictability of the rate of success are mandatory. The upscaling step requires an extensive knowledge of the biological processes underlying biodegradation, of the actors involved in and their interplay and of the chemical and physical factors contributing to reach the goal [131]. "Omics" approaches, by taxonomical and functional analyses of the data, offer a whole picture of the process with the identification of the key organisms within the microbiomes involved in the TPH degradation. At the same time, they offer an instrument to optimize the culturomic approach for the isolation of key actors of the whole process of decontamination, providing the instruments for the integration of culturomic and non-culturomic approaches, mandatory to boost the upscaling of the bioremediation technology and the number of sites that might be treated in the future. At the same time, the derived knowledge progressively increases the amount of the taxonomical and functional pool of genomic data for in silico interpretation of the microbiological characteristics of contaminated sites, eventually contaminated by vast arrays of contaminants. The further goal is the integration of analytical approaches based

on machine learning, such as the one described here for the analysis of omic-derived data, and the stocastic ones tipically adopted in the modelization of physical events, occuring in real situations, by a pipeline of analysis equipped with data that have been validated and calibrated with laboratory experiments and field cases. These latter are implemented in the design of processes on the real scale, since normally, providing instruments for the geotechnical and geological characterization and rheological engineering (e.g., MADflow, Available on line: http://madflow.ca, accessed on 1 March 2022). With reference to the extension and frequency of TPH contamination of soils all around the world and to the reach of already acquired data, the integration of all the analytical approaches will provide a robust instrument of prediction and control of the rate of success of a dedicated bio-based process.

**Author Contributions:** Conceptualization and investigation, I.C., S.B. and S.D.G.; writing—original draft preparation, I.C. and S.D.G.; writing—review and editing, I.C., S.B. and S.D.G.; supervision, S.D.G. All authors have read and agreed to the published version of the manuscript.

**Funding:** This research received no external funding.

**Data Availability Statement:** No new data were created or analyzed in this study. Data sharing is not applicable to this article.

**Conflicts of Interest:** The authors declare no conflict of interest.

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
