# Peer review of "Microbial Involvement in the Bioremediation of Total Petroleum Hydrocarbon Polluted Soils: Challenges and Perspectives"

_environments, doi:10.3390/environments9040052_

Round 1

Reviewer 1 Report

The article is interesting and well presented. It provides a lot of interesting information, although it is a bit long.

Author Response

REVIEWER: The article is interesting and well presented. It provides a lot of interesting information, although it is a bit long.

ANSWER: many thanks for your comments. With the help of all the reviewers the manuscript has been changed and improved in quality in its final form. We hope this version will encounter the your suggestions

Reviewer 2 Report

The paper provides a review devoted to the microbial involvement in the soil bioremediation. The problem is comprehensively considered in the review. However, there are some remarks to the paper.

Lines 76-79 should be deleted.

The authors should add more examples of the use of bacteria, fungi and interacting communities for soil bioremediation.

The conclusion should be more general. Now, the conclusion is more like a continuation of the review.

The authors should use more recent references, over the past 5 years.

The references should be written accordingly authors rules. Please check.

Lines 579, 608, 638, 682, 704, 736, 738, 747, 760, 763, 767, 777, 814, 827, 829, 846, 884, 889 – fix errors in references.

646, 698, 772, 823, 840, 852, 866 - incomplete references, complete them.

The authors should pay attention on the papers DOI 10.1007/s10653-019-00505-1, DOI 10.1007/s10653-019-00412-5, DOI 10.1016/j.apgeochem.2019.03.017.

104 – «…if time is working in their favor…» need to check style.

110 – «…to contrast their toxicity…» should be clarify.

383 – «…the decontamination of environmental matrices…» it might be better if you replace the word «matrices».

Author Response

REVIEWER COMMENTS: The paper provides a review devoted to the microbial involvement in the soil bioremediation. The problem is comprehensively considered in the review. However, there are some remarks to the paper.

MANY THANKS FOR YOUR COMMENTS THAT SIGNIFICANTLY IMPROVED THE MANISCRIPT IN THE PRESENT FORM

Lines 76-79 should be deleted.

IT HAS BEEN DELETED

The authors should add more examples of the use of bacteria, fungi and interacting communities for soil bioremediation.

EXAMPLES WERE ADDED

The conclusion should be more general. Now, the conclusion is more like a continuation of the review.

THE CONCLUSION HAS BEEN IMPROVED

The authors should use more recent references, over the past 5 years.

THE REFERENCES HAS BEEN IMPROVED

The references should be written accordingly authors rules. Please check.

CORRECTED

Lines 579, 608, 638, 682, 704, 736, 738, 747, 760, 763, 767, 777, 814, 827, 829, 846, 884, 889 – fix errors in references.

DONE

646, 698, 772, 823, 840, 852, 866 - incomplete references, complete them.

DONE

The authors should pay attention on the papers DOI 10.1007/s10653-019-00505-1, DOI 10.1007/s10653-019-00412-5, DOI 10.1016/j.apgeochem.2019.03.017.

THESE REFERENCES ARE VERY INTERESTING BUT WE THINK QUITE OUT OF THE DIECT SCOPE OF THIS REVIEW. MOREOVER, THE COMBINATION OF THE COMMENTS OF THE DIFFERENT REVIEWERS DID NOT PERMIT THE INCREMENT OF PROVIDED INFORMATION. WE HOPE THIS IS REASONABLE FOR THE REVIEWER 

104 – «…if time is working in their favor…» need to check style.

THIS PART HAS BEEN REPHRASED

110 – «…to contrast their toxicity…» should be clarify.

DONE

383 – «…the decontamination of environmental matrices…» it might be better if you replace the word «matrices».

DONE

Reviewer 3 Report

Dear authors,

Thank you very much for your interesting summary in this problematic. The topic of this paper is very important and knowing the methods and finding new perspectives are key in bioremediation processes of risk elements. Before the paper can be accepted for publication, I have some comments and questions that might improve the quality of the paper.

  1. Please, check the English, as there are some grammatical mistakes
  2. I would recommend to slightly rewrite the abstract and make it easier and more readable
  3. In the introduction part (maybe after introduction) I would add the information about the impact of crude oil/TPH on environment components in general and living organisms (include plants and animals)
  4. Line 294 – The whole story in this part was about the fungi and there you jump to the quorum sensing bacterial communication. I am not sure if it this was the right step and you should rather to explain why you are going to tell about it. Just add some sentence to connect these ideas.

Wish you good luck with your future studies.

Author Response

Thanks for your comments and suggestion

REVIEWER COMMENTS:

  1. Please, check the English, as there are some grammatical mistakes

THE ENGLISH HAS BEEN REVISED

  1. I would recommend to slightly rewrite the abstract and make it easier and more readable

THE ABSTRACT HAS BEEN REVISED

  1. In the introduction part (maybe after introduction) I would add the information about the impact of crude oil/TPH on environment components in general and living organisms (include plants and animals)

THE COMBINATION OF THE COMMENTS OF THE DIFFERENT REVIEWERS DID NOT PERMIT THE INCREMENT OF PROVIDED INFORMATION. ANYWAY WE THINK THAT MUCH HAS BEEN DONE OF TPH AND MICH IS ALREADY KNOWN AROUND THIS ASPECT THAT CAN BE OMITTED AT THIS STAGE, WE HOPE THIS IS REASONABLE FOR THE REVIEWER 

  1. Line 294 – The whole story in this part was about the fungi and there you jump to the quorum sensing bacterial communication. I am not sure if it this was the right step and you should rather to explain why you are going to tell about it. Just add some sentence to connect these ideas.

THE PARAGRHAPH HAS BEEN IMPROVED, THANKS FOR YOUR OBSERVATION

Round 2

Reviewer 2 Report

The authors considered all my comments. The manuscript can be accepted in its present form and is worthy of publication in Environments journal.